# Key Genes of Immunity Associated with Pterygium and Primary Sjögren’s Syndrome

**DOI:** 10.3390/ijms24032047

**Published:** 2023-01-20

**Authors:** Yumeilan Liu, Hao Chen, Hongping Cui

**Affiliations:** 1Department of Ophthalmology, Shanghai East Hospital, Tongji University School of Medicine, Shanghai 200120, China; 2Department of Biochemistry and Molecular Biology, Tongji University School of Medicine, Shanghai 200120, China

**Keywords:** pterygium, primary Sjögren’s Syndrome, bioinformatics analysis, immune response

## Abstract

Pterygium and primary Sjögren’s Syndrome (pSS) share many similarities in clinical symptoms and ocular pathophysiological changes, but their etiology is unclear. To identify the potential genes and pathways related to immunity, two published datasets, GSE2513 containing pterygium information and GSE176510 containing pSS information, were selected from the Gene Expression Omnibus (GEO) database. Differentially expressed genes (DEGs) of pterygium or pSS patients compared with healthy control conjunctiva, and the common DEGs between them were analyzed. Gene Ontology (GO) and Kyoto Encyclopedia of Genes and Genomes (KEGG) enrichment analysis were conducted for common DEGs. The protein–protein interaction (PPI) network was constructed using the STRING database to find the hub genes, which were verified in clinical samples. There were 14 co-upregulated DEGs. The GO and KEGG analyses showed that these common DEGs were enriched in pathways correlated with virus infection, antigen processing and presentation, nuclear factor-kappa B (NF-κB) and Th17 cell differentiation. The hub genes (*IL1R1*, *ICAM1*, *IRAK1*, *S100A9*, and *S100A8*) were selected by PPI construction. In the era of the COVID-19 epidemic, the relationship between virus infection, vaccination, and the incidence of pSS and pterygium growth deserves more attention.

## 1. Introduction

Pterygium is a fibrovascular-like tissue that connects to the conjunctiva and grows towards the surface of the cornea [1,2,3,4], which causes corneal astigmatism, blockage of the visual axis, and eventually causes visual impairment [5,6]. Due to population and regional differences, the incidence of pterygium presence ranges from 2.3% to 58.8% [7,8,9,10,11]. Currently, surgical resection is the most commonly used clinical treatment, but postoperative discomfort and high recurrence rate are very serious issues [12,13]. The pathological mechanism is still not completely understood. Previous epidemiologic studies suggested that geographical location, exposure time to sunlight and sand, dry eyes, type I allergy, and human papillomavirus infection were risk factors for the occurrence and progress of pterygium growth [10,11]. Abnormal tear function has been observed in patients with pterygium, including higher tear osmolarity, decreased percentage of crystals, and lower goblet cell density [14]. In addition, a long-term postoperative inflammatory state caused by dry eye can aggravate the recurrence rate of pterygium, while using artificial tears after excision could lower the recurrence rate [15,16]. Previous studies have shown a potential association between pterygium development and dysfunction of the ocular surface.

Primary Sjögren’s syndrome is an autoimmune disease characterized by dry mouth and dry eyes, which remains of unknown specific etiology. Lymphocyte foci in exocrine glands can be seen on histological examination [17]. The prevalence of pSS in adults is about 5%, with a male-to-female ratio of 1:9, and is especially higher in Asian women [18,19]. Ocular pathophysiological changes, such as aqueous-deficient dry eyes, meibomian gland dropout, and mucin deficiency, are often associated with pSS [20,21,22]. Compared with dry eye but non-pSS patients, the ocular surface symptoms of pSS patients are more severe, and even complications such as corneal perforation and scleritis may occur [23]. The exact etiology and pathogenesis of pSS are also unknown. However, there are known associations between autoimmune diseases and genetic variants in the human leukocyte antigen (HLA) region [24]. Antigen presentation involving T-cells and activation of type I interferon (IFN-1) signaling pathway play an important role in pSS pathogenesis [25].

As described above, a pterygium and pSS are both related to ocular surface dysfunction and eye inflammation, but their commonalities in etiology are lacking in recognition [10,11,14,15,16,17,20,21,22]. Bioinformatics analysis of DEGs helps to discover key genes and pathways in diseases. Yuting Xu’s research revealed the competing endogenous RNA (ceRNA) regulation mechanisms during pterygium pathogenesis. The lncRNA LIN00472-dominated ceRNA network containing multiple miRNAs and downstream target genes participates in the pathological processes, such as abnormal cell adhesion and proliferation of the pterygium, through the PID/FOXM1 pathway [26]. Siying He’s study found that several differentially expressed miRNAs and DEGs, especially the miR-29-3p and collagen family genes, were involved in regulating cell death, extracellular matrix breakdown, and the EMT process of the pterygium [27]. Naoko’s study showed that UV exposure promotes DEGs expression in the pterygium [28]. Recent studies show that DEGs in pSS were enriched in viral infection, activation of immune cells, and mitochondrial metabolism-related signaling pathways. [29,30]. All in all, there have been many studies on the bioinformatics analysis of pterygium growth and pSS, but the analysis of the correlation between these two diseases is lacking.

In this research, we worked for a better knowledge about the pathogenesis at the genetic level between pterygium growth and pSS by analyzing data from the GEO database. This is the first report of common DEGs for the two clinically relevant diseases, pterygium growth and pSS. The flow chart of this research is shown in Figure 1. First, through the datasets downloaded from the GEO database, we identified the DEGs of pterygium growth or pSS compared with healthy control conjunctiva and screened out common genes. Then hub genes and related pathways were obtained through GO and KEGG enrichment analysis and PPI network construction. Consequently, the hub genes (*IL1R1*, *ICAM1*, *IRAK1*, *S100A9*, and *S100A8*), and immune response to viral infection, IL-1 and S100A8/A9 related signaling pathways were selected.

## 2. Results

### 2.1. Identification of DEGs and Intersection Set in Pterygium Growth and pSS

In the pterygium dataset GSE2513, 1601 DEGs were screened out when we compared the eight pterygium samples with four healthy controls. A total of 906 DEGs were up-regulated, while 695 DEGs were down-regulated (*p* < 0.05). In the pSS dataset GSE176510, 147 immune pathway-related DEGs were screened out when we compared the seven pSS samples with 19 healthy controls. A total of 137 DEGs were up-regulated, while 10 DEGs were down-regulated (*p* < 0.05). The top DEGs expression profiles of pterygium and pSS samples are presented by volcano plots and heatmaps (Figure 2). After analysis using the online analysis tool Venn, 14 genes (*IRAK1, LEF1, CTSS, HLADPA1, S100A8, ARHGDIB, CD59, TAP2, ICAM1, IL4R, CEACAM1, BAX, IL1R1, S100A9*) that were up-regulated in both pterygium and pSS samples were screened out, while no DEGs were commonly down-regulated (Figure 3). The selected 14 commonly up-regulated DEGs were used for subsequent functional analysis.

### 2.2. GO and KEGG Enrichment Pathway Analysis

Functional enrichment and pathway analyses of 14 commonly up-regulated DEGs in pterygium and pSS samples were performed at the threshold of *p* < 0.05 (Figure 4 and Figure 5). Changes in GO biological processes (BP) mainly included immune response and cell-cell adhesion (e.g., T-cell activation, regulation of cell-cell adhesion, leukocyte cell-cell adhesion, and neutrophil activation and degranulation). Changes in cellular component (CC) were notably focused on enrichment of cell outer membranes, such as collagen-containing extracellular matrix, tertiary granule, transport vesicle and external side of plasma membrane. Moreover, in the molecular function (MF) section, changes were significantly occurred in receptor activity (RAGE receptor binding, Toll-like receptor binding, and cytokine receptor binding), and binding-related function (fatty acid binding and heat shock protein binding). In particular, the KEGG analysis indicated that changes in signaling pathways were mostly enriched in virus infection (Epstein–Barr virus infection, Human T-cell leukemia virus 1 infection, Herpes simplex virus 1 infection), antigen processing and presentation, and immune-related pathways (NF-κB signaling pathway, Th17 cell differentiation and phagosome).

### 2.3. PPI Network Analysis and Hub Gene Selection

To discriminate the hub genes from the 14 commonly up-regulated DEGs in pterygium and pSS samples, a PPI network was constructed. Interleukin 1 receptor type I (IL1R1), intercellular adhesion molecule 1 (ICAM1), interleukin-1 receptor-associated kinase 1 (IRAK1), S100 calcium binding protein A9 (S100A9) and S100 calcium binding protein A8 (S100A8) showed comparatively higher degrees in the PPI network and were discriminated as hub genes. Furthermore, the PPI network was divided into three clusters centered on IL1R1 and S100 proteins A8/9 (Figure 6).

### 2.4. Hub Genes Expression in Clinical Samples

There was no significant difference in gender and age among the three groups of clinical samples (Table 1). Compared with the healthy control conjunctiva, the expression of five hub genes (*IL1R1*, *ICAM1*, *IRAK1*, *S100A9*, and *S100A8*) in pterygium or pSS samples was statistically significant (*p* < 0.05) (Figure 7).

## 3. Discussion

In the current study, gene expression analysis was performed on previously published datasets of pterygium and pSS patients to uncover shared gene signatures underlying their clinical relevance. Several DEGs in pterygium and pSS tissues were identified from the datasets, and five hub genes (*IL1R1*, *ICAM1*, *IRAK1*, *S100A9*, and *S100A8*) were finally screened out. Bioinformatics analyses show the common DEGs are remarkably enriched in pathways associated with virus infection, antigen processing and presentation, NF-κB signaling pathway, Th17 cell differentiation, and neurotrophin signal transduction. Further functional investigations of these genes and pathways are necessary to elucidate their roles in the pathogenesis of pterygium and pSS occurrence.

The IL-1 family (IL-1F) is a group of highly structurally conserved exocrine cytokines involved in diverse immune responses [31,32]. Among the 11 members of this family, IL-1α and IL-1β are the most studied, the former is widely expressed in various cells, and the latter is mainly secreted by monocyte-macrophages [33]. The receptor for IL-1F consists of 10 transmembrane proteins with a similar structure, which is formed by three Ig-like domains responsible for ligand binding, a transmembrane domain, and an intracellular portion with the Toll-IL-1-receptor (TIR) domain, responsible for signal transduction. The IL-1F binds to co-receptors to form a ligand-receptor complex, which mediates interleukin-1-dependent activation of NF-κB, MAPK, and other pathways by recruiting TOLLIP, MYD88, IRAK1 or IRAK2 and other receptor proteins [34,35,36]. Previous studies have shown that, compared with healthy controls, the expression of IL-1α and IL-1β is obviously increased in the lacrimal and salivary gland tissues of patients with pSS [37,38,39]. In addition, the high expression of IL-1F in pSS patients could recruit T-lymphocytes and cause long-term inflammatory response, which led to ocular surface squamous metaplasia [40]. Previous study also shows that IL-1β promotes the matrix metallopeptidase-9 (MMP-9) production and migration of pterygium fibroblasts [41]. Blockers of IL1RI and IL-1β have been used in the treatment of rheumatoid arthritis, breast cancer, and other diseases, but there is still a lack of research and application in ocular surface inflammatory diseases [42,43,44].

The S100 proteins are calcium-binding proteins that can combine with Ca^2+^ and other metal ions to exert intracellular activity and regulate calcium homeostasis, cell cycle, and cell growth. The S100 proteins can also bind to receptors such as advanced glycation end products and Toll-like receptors through paracrine for extracellular regulation. These processes can lead to activation of T-cells and release of inflammatory factors, thereby damaging the immune homeostasis of the conjunctiva and causing ocular surface inflammation [45,46,47]. Protein and gene level testing confirmed that the expression level of S100A8/9 proteins in pterygium tissue was higher than that in normal conjunctiva tissue. The highly expressed S100A8/9 protein can bind to the keratin filament expressed during terminal differentiation, which may promote the reorganization of cytoskeleton in the process of pterygium hyperproliferation [48]. Compared with primary pterygium tissue, S100A7 expression was increased in recurrent pterygium tissue and MAPK inflammatory response pathway was activated [49]. Proteomic Profiling showed that S100A9, Histone H1.4, and neutrophil collagenase were upregulated in saliva and tears of pSS patients [50]. In the pSS rabbit eye model, S100A6 and S100A9 proteins in tears were up-regulated, while the polymeric immunoglobulin receiver (pIgR), and immunoglobulin gamma chain C region were downregulated [51]. These phenomena are inseparable from the function of S100 proteins in activating the innate immune system and altering the immune tolerance of the eye. In the corneal neovascularization animal model, the expression levels of S100 proteins, especially S100A8/9, were significantly increased, and subconjunctival injection of antibodies could inhibit angiogenesis [52]. Antibody treatment with anti-S100A8/9 has been proven effective in mouse models of colitis, acute myocardial infarction, and pancreatitis, which reveals the effect of anti-S100A8/9 antibody treatment on inflammatory diseases [53,54,55]. Although direct study of pterygium or pSS syndromes with anti-S100A8/9 is lacking, the treatment of other diseases effectively reduces a variety of inflammatory factors, which are also involved in ocular immunity, thus suggesting the feasibility of anti-S100A8/9 in the treatment of pterygium or pSS syndromes [47].

Our research revealed that common DEGs in pSS and pterygium disease were enriched in pathways associated with virus infection, antigen processing, and presentation. The microbiota, including bacteria, fungi, viruses, protozoa, and eukaryotes, contributes to maintaining ocular surface homeostasis and immune tolerance, but can be destroyed by infection [56]. Persistence of human papillomavirus (HPV) infection was found to be correlated with postoperative pterygium recurrence [57]. Recent research provides strong serological evidence for the association among Epstein–Barr virus (EBV), human T-cell leukemia virus type 1 (HTLV-1) infection and pSS [58,59,60]. The coronavirus disease 2019 (COVID-19) pandemic is posing a serious threat to global public health. Viral infections and ocular surface diseases deserve more attention. Di Ma’s research revealed that pterygium tissue had higher expression of SARS-CoV-2 receptors ACE2 and TMPRSS2 than normal conjunctiva in the mouse model [61]. Many studies have shown that patients with pSS are more susceptible to COVID-19 infection, and the type I interferon pathway may be a common mechanism [62]. It has also been suggested that COVID-19 vaccination may lead to early onset of subclinical pSS, but long-term validation with large sample sizes is lacking [63]. At present, sustained antiviral ocular drug delivery systems including nanocarriers, prodrugs and in situ gels are receiving extensive attention. However, due to the existence of ocular barrier and the lack of specific drugs for viruses, further research is still required.

The pathogenesis of pterygium growth remains unknown. Some hypotheses suggested that the occurrence and development of a pterygium were related to the destruction of limbal stem cell barrier, cell aging, and epithelial–mesenchymal transition (EMT). However, our analysis shows that gene expression related to virus infection and immunity may play an important role in the pathogenesis of a pterygium and pSS. Although the data from the GEO database are insufficient, qPCR detection of clinical samples confirmed that these hub genes really have significant differences in expression in pSS conjunctiva and in a pterygium, which deserves more attention. This is the first analysis of common DEGs for these two clinically relevant diseases. Literature studies suggest that IL-1, S100A8/9, viral infection, and the above-involved signal molecules could be used as pathogenesis, prognosis predictors and potential therapeutic targets of pterygium growth and pSS, but further validation experiments are needed.

## 4. Materials and Methods

### 4.1. Microarray Data Collection

The datasets were obtained from the GEO database using “pterygium”, “pterygia” and “Sjögren’s syndrome” as search keywords. We established the following screening criteria: (1) The dataset is *Homo sapiens* data. (2) The database contains sequencing information of total RNA extracted from conjunctiva of patients and healthy controls. (3) The dataset has no missing data. (4) The original data of the dataset can be downloaded. Finally, we chose the GSE2513 and GSE176510 datasets. The GSE2513 dataset contained the gene expression profiles of eight pterygium patients and four healthy conjunctiva controls and was analyzed using the GPL96 [HG-U133A] Affymetrix Human Genome U133A Array [64,65,66]. The GSE176510 dataset contained the expression profiles of immune pathway-related genes of seven pSS patients and 19 age-and-sex-matched healthy controls and was analyzed using the GPL28577 NanoString Human Immunology v2 Code Set.

Information about the datasets GSE2513 and GSE176510 is available in [Gene Expression Omnibus (GEO) database] at [https://www.ncbi.nlm.nih.gov/gds (accessed on 17 March 2022)].

### 4.2. Data Processing and Identification of DEGs

All original data were downloaded from the GEO database and standardized using the limma package of R 3.2.3. Probes were annotated according to annotation files and would be removed without corresponding gene symbols. When multiple probes matched the same gene symbol, the average value was calculated for subsequent experiments. The DEGs between pterygium patients and healthy controls were screened from the GSE2513 dataset using the classical Bayesian method with a threshold of *p* < 0.05. The DEGs between the pSS patients and healthy controls based on the GSE176510 were screened using the same method. Then we used the online diagram tool (http://bioinformatics.psb.ugent.be/webtools/Venn/ (accessed on 20 March 2022)) to construct a Venn diagram of the DEGs screened from the two datasets to get their intersection set, which were used for subsequent analysis. To visualize the DEGs, R 3.2.3 and a free online application (http://www.heatmapper.ca/expression/ (accessed on 20 March 2022)) was used to construct volcano plots and heatmaps.

### 4.3. Functional Enrichment Analyses for Intersection DEGs

To evaluate relevant biological functions and signaling pathways, GO and KEGG pathway enrichment analyses were carried out using the Database for Annotation Visualization and Integrated Discovery (DAVID) (https://david.ncifcrf.gov/home.jsp (accessed on 20 March 2022)) for intersecting common DEGs, with the threshold of *p*-adjusted < 0.05 [67,68]. The Benjamini–Hochberg method was used to correct the *p*-value.

### 4.4. Construction of PPI Network and Identification of Hub Genes

Using the protein interaction network to trace the upstream and downstream relationships of signal transmission and gene expression regulation, the key genes and functional modules involved in disease occurrence and progression can be effectively identified. To further explore the interactions among the common DEGs of these two clinically correlated diseases, a PPI network was established through the STRING database (https://cn.string-db.org/ (accessed on 20 March 2022)), with the organism as “Homo sapiens” [69]. Moderate confidence (0.400) was set as the minimum required interaction score to ensure statistical significance. The clustering option of the PPI network was kmeans clustering. In the network exported pictures, the nodes represent the proteins, while the lines represent the interactions between proteins. Then, the Cyto-Hubba plugin of Cytoscape 3.7.2 was used to examine the PPI network and nominate the hub genes with top node degrees.

### 4.5. Acquisition of Clinical Samples

In order to verify the hub DEGs screened from the database, we selected three pterygium patients, three pSS patients, and three healthy controls for analysis. The experiment was conducted according to ethical requirements. All subjects signed an informed consent form. Each participate with a pterygium or pSS met the following criteria: (1) The patient was definitely diagnosed as having a pterygium or pSS. (2) The patient had no other eye diseases. (3) The patient had no major systemic disease. (4) Randomized gender of patients. (5) The patient was 30–80 years old. The basis for the inclusion of healthy controls was the same as the above-mentioned criteria, but all eye diseases were excluded. Patients with a pterygium volunteered to undergo pterygium resection, and pterygium tissue was collected during the operation. Conjunctival cells were obtained from pSS and healthy controls by impression cytology method. All acquired organizations were immediately stored at −80 °C.

### 4.6. Tissue RNA Extraction and Quantitative Real-Time PCR (qPCR)

Total RNA was extracted from the frozen tissues with RNAiso Plus reagent (Takara, Dalian, China), and then reverse transcribed using PrimeScript™ RT Master Mix reagent (Takara, Dalian, China). The qPCR experiment was conducted using SuperReal PreMix Plus (SYBR Green) reagent (Tiangen, Beijing, China) in the CFX96™ Real-Time PCR Detection system (Bio-Rad, Hercules, CA, USA). The gene GAPDH was selected as the internal reference gene. The expression of all genes was measured by Ct value and normalized with the healthy control group. Primers used for qPCR detection come from the data published by PrimerBank (Table 2).

### 4.7. Statistical Analysis

For comparison of two groups of quantitative data, unpaired *t*-tests were conducted in GraphPad Prism program. For comparison of multiple groups of quantitative data, one-way ANOVA analysis was used. The mean of each test group was compared with the mean of control group. For comparison of qualitative data, chi-square test was conducted in SPSS program. Statistical significance was regarded as *p* < 0.05.

## 5. Conclusions

We identified a shared gene signature (*IL1R1*, *ICAM1*, *IRAK1*, *S100A9*, and *S100A8*) between pterygium presence and pSS. The analysis of this signature underlined that immune response to viral infection, IL-1, and S100A8/A9 related signaling pathways probably play vital roles in the development of these two clinically correlated diseases. This finding may help in the identification of new therapeutic targets and understanding of the pathological mechanisms.

## Figures and Tables

**Figure 1 ijms-24-02047-f001:**
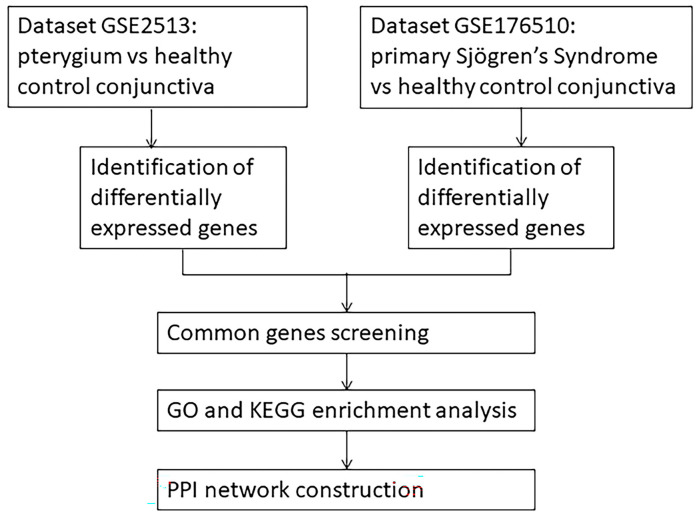
Workflow of this study. Dataset GSE2513 and GSE176510 was previously published in [Gene Expression Omnibus (GEO) database] at [https://www.ncbi.nlm.nih.gov/gds (accessed on 17 March 2020)]. GO: Gene Ontology. KEGG: Kyoto Encyclopedia of Genes and Genomes.

**Figure 2 ijms-24-02047-f002:**
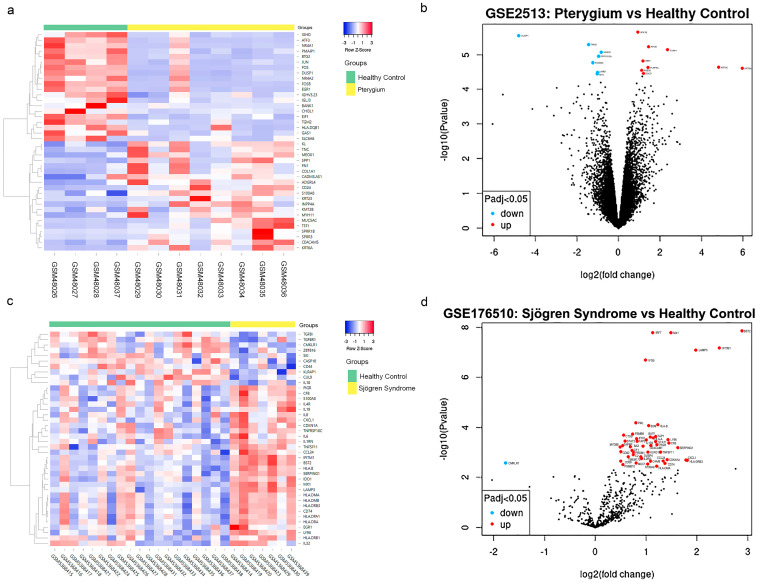
Identification of differentially expressed genes (DEGs) in pterygium and primary Sjögren Syndrome (pSS) samples. (**a**) Heatmap and (**b**) volcano plot of the DEGs between samples from pterygium and healthy control conjunctiva (GSE2513). (**c**) Heatmap and (**d**) volcano plot of the DEGs between samples from pSS and healthy control conjunctiva (GSE176510). Information about the dataset GSE2513 and GSE176510 is available in [Gene Expression Omnibus (GEO) database] at [https://www.ncbi.nlm.nih.gov/gds (accessed on 17 March 2022)]. DEGs between patients and healthy control conjunctiva were screened from the dataset using the classical Bayesian method with a threshold of *p* < 0.05.

**Figure 3 ijms-24-02047-f003:**
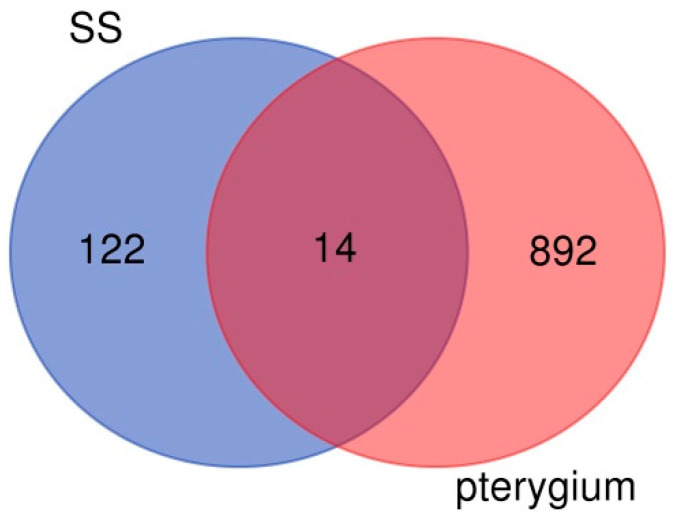
Venn diagram of the up-regulated differentially expressed genes (DEGs) in primary Sjögren Syndrome (pSS) and pterygium samples. Left circle represents pSS, while right circle represents pterygium samples. 14 genes (*IRAK1*, *LEF1*, *CTSS*, *HLADPA1*, *S100A8*, *ARHGDIB*, *CD59*, *TAP2*, *ICAM1*, *IL4R*, *CEACAM1*, *BAX*, *IL1R1*, *S100A9*) that are up-regulated in both pterygium and pSS samples were screened out. The online diagram tool [http://bioinformatics.psb.ugent.be/webtools/Venn/ (accessed on 20 March 2022)] was used to construct a Venn diagram.

**Figure 4 ijms-24-02047-f004:**
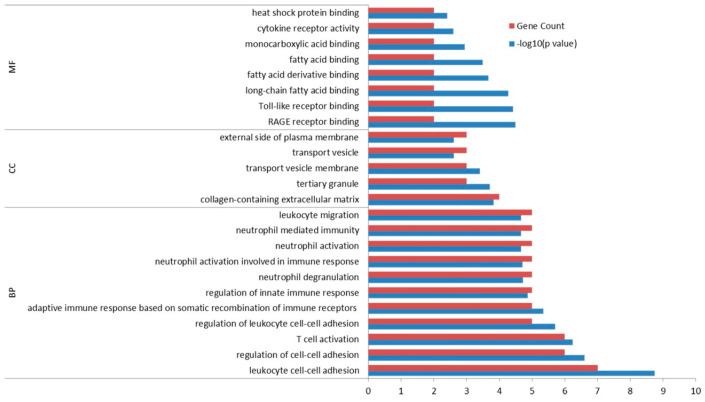
GO enrichment analysis of the common 14 up-regulated differentially expressed genes (DEGs) in both pterygium and primary Sjögren Syndrome (pSS) samples. MF: molecular function. BP: biological processes. CC: cellular component. The red bar represents the number of DEGs corresponding to the GO analysis term. Blue bar represents −log10 (*p* value). (*p* < 0.05).

**Figure 5 ijms-24-02047-f005:**
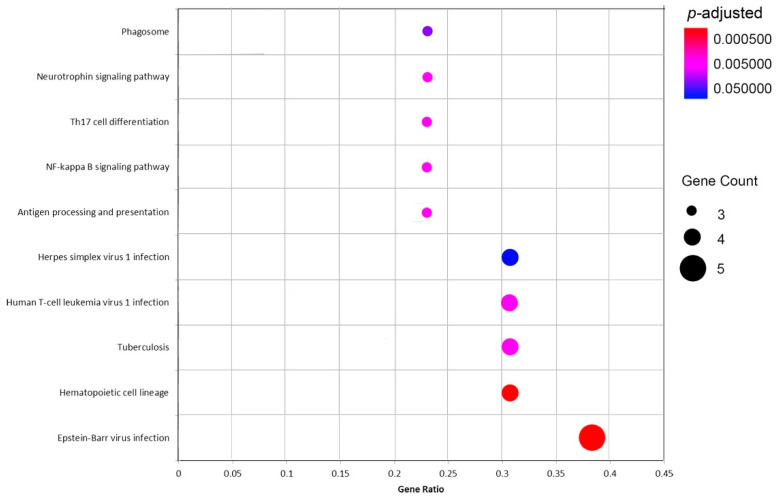
KEGG enrichment analysis of the common 14 up-regulated differentially expressed genes (DEGs) in both pterygium and primary Sjögren Syndrome (pSS) samples. Gene Ratio = The number of genes enriched on this term/the total number of input DEGs. The size of the circle represents the number of DEGs, and the color represents the value of *p*-adjusted.

**Figure 6 ijms-24-02047-f006:**
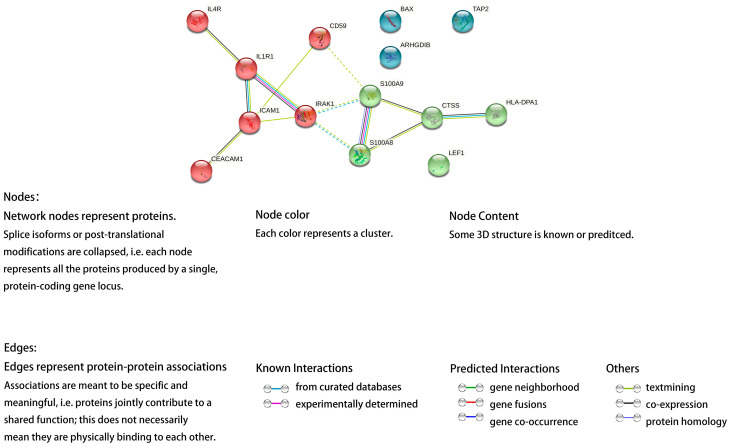
PPI network of the common 14 up-regulated differentially expressed genes (DEGs) in both pterygium and primary Sjögren Syndrome (pSS) samples.

**Figure 7 ijms-24-02047-f007:**
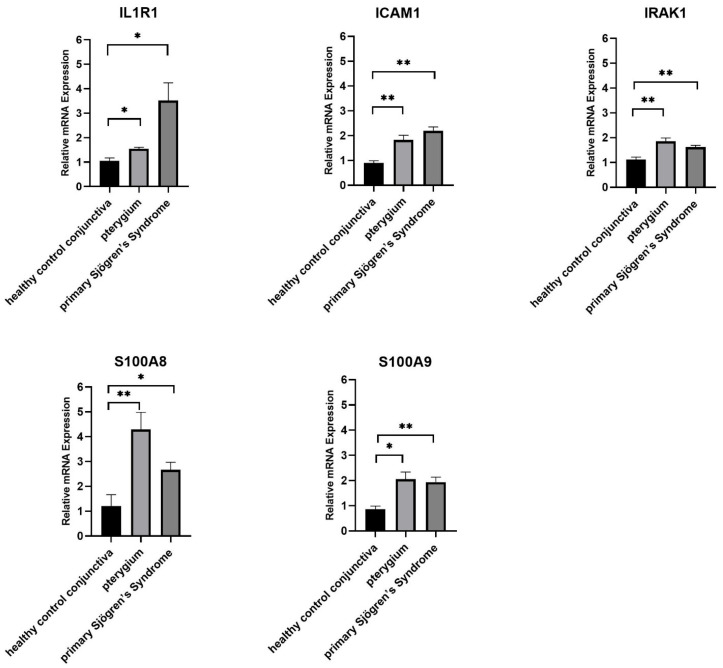
Hub genes expression in clinical samples. The expression of all genes was measured by Ct value and normalized with the healthy control group. Brown–Forsythe and Welch ANOVA tests were used for comparison. *: *p* < 0.05. **: *p* < 0.01.

**Table 1 ijms-24-02047-t001:** Clinical data of the participants.

Participant Number	Group	Sex	Age
1	healthy control	Male	45
2	healthy control	Male	58
3	healthy control	Female	75
4	pterygium	Female	59
5	pterygium	Male	72
6	pterygium	Female	77
7	primary Sjögren Syndrome	Female	66
8	primary Sjögren Syndrome	Female	68
9	primary Sjögren Syndrome	Male	57

**Table 2 ijms-24-02047-t002:** Primers used for qPCR detection.

Gene	Forward Primer	Reverse Primer
*GAPDH*	GGAGCGAGATCCCTCCAAAAT	GGCTGTTGTCATACTTCTCATGG
*IL1R1*	ATGAAATTGATGTTCGTCCCTGT	ACCACGCAATAGTAATGTCCTG
*ICAM1*	GTATGAACTGAGCAATGTGCAAG	GTTCCACCCGTTCTGGAGTC
*IRAK1*	GCACCCACAACTTCTCGGAG	CACCGTGTTCCTCATCACCG
*S100A9*	GGTCATAGAACACATCATGGAGG	GGCCTGGCTTATGGTGGTG
*S100A8*	ATGCCGTCTACAGGGATGAC	ACTGAGGACACTCGGTCTCTA
*GAPDH*	GGAGCGAGATCCCTCCAAAAT	GGCTGTTGTCATACTTCTCATGG
*IL1R1*	ATGAAATTGATGTTCGTCCCTGT	ACCACGCAATAGTAATGTCCTG
*ICAM1*	GTATGAACTGAGCAATGTGCAAG	GTTCCACCCGTTCTGGAGTC

## Data Availability

Information about the dataset GSE2513 and GSE176510, which were used in this study, is available in [Gene Expression Omnibus (GEO) database] at [https://www.ncbi.nlm.nih.gov/gds (accessed on 17 March 2022)].

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
