# Peer review of "Key Genes of Immunity Associated with Pterygium and Primary Sjögren’s Syndrome"

_ijms, 2023, doi:10.3390/ijms24032047_

Round 1
Reviewer 2 Report
the paper is interesting and well-written. The material and methods are adequate. The Authors should add some information regarding the involvement of S100 protein in the regulation of eye homeostasis particularly in inflammation. The data should add evidence regarding the potential therapeutic role of 100 protein in the analysed disorders.
Round 2
Reviewer 1 Report
The authors have addressed each of the original concerns, which were mostly clarifications.
Reviewer 2 Report
The Authors have added the requested information. The manuscript is now suitable for publication in IJMS.